# Promoting physical activity and academic achievement through physically active learning: Qualitative perspectives of co-design and implementation processes

Laurie Simard[1,2,3]*, Julie Bouchard[1,2], Martin Lavallière[1,2], Tommy Chevrette[1,3]

1 Centre Intersectoriel en Santé Durable, Université du Québec à Chicoutimi, Saguenay, Canada,
2 Laboratoire BioNR, Université du Québec à Chicoutimi, Saguenay, Canada, 3 Observatoire du Développement Moteur et Psychomoteurs des 0–18 ans, Université du Québec à Chicoutimi, Saguenay, Canada

* laurie_simard@uqac.ca

## Abstract

This article discussed the issue of low PA levels among school-aged children and highlights the promising approach of school-based interventions, including physically active learning (PAL), to increase PA levels. The study aimed to co-design and to assess the implementation of a PAL program for 8 weeks in 4 elementary classrooms (82 students and 7 teachers), emphasizing the potential of integrating PA with academic learning and the importance of co-designing programs with teachers to maximize their effectiveness. Technology was found to support PAL practices in promoting PA and academic achievement. The study underscores the need for further research to explore the societal implications of PAL programs, including their potential to improve the health and well-being of children while promoting positive academic outcomes.

## Introduction

Physical activity (PA) is essential for the optimal development and well-being of school-aged children (5 to 17 years old). A strong body of evidence has linked PA to a wide range of health outcomes, including reduced adiposity, improved physical fitness, cognitive function, and mental health [1]. Despite the numerous benefits of PA, a significant proportion (67%) of Canadian school-aged children fail to meet the recommended PA levels of at least 60 minutes per day doing moderate- to vigorous-intensity aerobic PA [2, 3], highlighting the need for effective interventions to increase PA levels in this population.

School-based PA programs, particularly those involving physically active learning (PAL), have been proposed as a promising approach to increase PA levels in children [4–10]. Moreover, PAL interventions have shown promising results in improving academic achievement [10–13], student enjoyment [12], time on task [8, 10, 13, 14], and classroom behaviour [15].

**Data Availability Statement:** Data cannot be shared publicly because of participants have agreed to deposit the research data exclusively in

the database of the Observatoire du développement moteur et psychomoteur des 0-18 an. Data are available from this Institutional Data Access / Ethics Committee (email address observatoire_dmp@uqac.ca).

**Funding:** This research was supported by grants from the Syndicat des chargées et chargés de cours de l'Université du Québec à Chicoutimi (SCCCUQAC) to L.S., the Social Sciences and Humanities Research Council of Canada (SSHRC) to L.S. and the Regional Education Research Consortium to L.S. and T.C The funders had no role in study design, data collection and analysis, decision to publish, or preparation of the manuscript.

**Competing interests:** Cogni-Actif, the PAL program, has been co-designed in an action research project led by the authors. No other potential conflict of interest was reported by the authors.

School-based PA programs, particularly those involving physically active learning (PAL), have been proposed as a promising approach to increase PA levels in children [12, 16, 17].

## Objective

To address these challenges, this study aimed to co-design a PAL program and assess its implementation in elementary schools, focusing on teachers' and students' perceptions. The co-design process involved collaboration between researchers and teachers to tailor the program to their needs and preferences. The study findings could help overcome the challenges faced in implementing PAL interventions, leading to the successful design, development, and implementation of future programs. These results have important implications for developing effective school-based PA programs, improving the health and well-being of school-aged children, and promoting their academic achievement.

## Theory

To ensure that the PAL program developed for elementary school students is effective, feasible, and consistent with the school's curriculum and mission [17], a co-design process has been prioritized as the underlying theory guiding the methodology [18]. This iterative, exploratory, and cyclical process involves collaboration between school practitioners, such as teachers, health and physical education teachers, and pedagogical advisors, and an interdisciplinary research team consisting of experts in kinesiology, neuropsychology, and digital art.

The theoretical foundations of the PAL program are based on key elements of three previously published and validated school-based PA programs: "Take 10!", "FUNterval" and "Fit en Vaardig op school". The Take 10! program from the United States [19] combines academic instruction with 10 minutes of PA. This program was designed with teachers and education experts and includes specific learning objectives for each topic of academic content. Materials such as characters, posters, cards, and rewards are designed to be attractive, user-friendly, and sustainable for children from 5 to 11 years old. The program is effective in raising children's PA levels and reducing classroom inattentive behaviour and is easily implemented from kindergarten through fifth grade [10, 20, 21].

The FUNterval program from Canada [22] is a PA breaks program without academic content designed for third to fifth grade (8 to 11 years old). It relies on high-intensity interval activities that take only 4 minutes to complete, consisting of 20 seconds of high-intensity activity separated by 10 seconds of rest, repeated 8 times. Interactive storylines are used to motivate children to engage in those activities, which require no equipment and are designed to be accomplished in a classroom. The short duration of the program, its sequences of moderate and vigorous intensity PA, and its effectiveness in improving students' selective attention make it an interesting insight for the design of a PAL program [22].

The Fit en Vaardig op school program from Netherlands [23] includes 63 physically active academic classroom lessons for second and third grade (7 to 9 years old). Each lesson involves math and language problems performed while engaging in moderate- to vigorous-intensity PA behind or beside the school desk, supported by a presentation on the interactive whiteboard. This intervention resulted in improved math and reading scores. Teachers involved in this study provided sound advice for improving future PAL interventions, indicating that intensive movements should not be paired with challenging academic content to keep students engaged [23].

By drawing on the key elements of these three successful programs, the theoretical foundations of the PAL program are well-established. The co-design process involving school practitioners and an interdisciplinary research team is intended to ensure that the program is tailored

to the specific needs of the students and aligns with the school's curriculum and mission [24, 25]. The use of evidence-based programs and ministerial referents, the collaboration between school practitioners and research experts, and the consideration of specific school and student needs are expected to maximize the effectiveness and sustainability of the PAL program.

## Material and methods

### Research design

This study employed an embedded experimental design, a mixed-method approach that concurrently incorporated a qualitative component into a quantitative experimental design [26]. This article focuses on the qualitative data documenting the co-design and implementation processes as reported by teachers and students. The quantitative data on the assessment of the PAL will be discussed in a separate article.

### Ethics

The University du Québec à Chicoutimi's Research and Ethics Committee (REC-UQAC) approved this research (CER 2020–393). Prior to initiating the co-design of the PAL program, electronic written informed consent was obtained from school practitioners (February 2020), children and parents/guardians before the implementation of the PAL program (October 2020). The study adhered to public health guidelines related to the COVID-19 pandemic, with permission from the University COVID-19 Committee and the Lac-Saint-Jean School Service Centre (Québec, Canada). Children and parents were informed of these preventive measures in the informed consent form.

### Sampling, recruitment, and participants

From the start to the end date, his study involved seven teachers (N = 7): four fourth-grade teachers (4GTs, n = 4) and three health and physical education teachers (HPETs, n = 3) from three different schools, in addition to the research team. The PAL program was implemented in the classrooms of the 4GTs involved. Of the 99 students who participated in the PAL program, 83 gave informed consent. One participant changed schools before data collection and was excluded, resulting in 82 students (boys, n = 37; girls, n = 45; aged 9.8±0.4 years) included in the final data analysis (see Table 1 for a breakdown by classroom). Information about age and gender was obtained from an electronic questionnaire completed by parents at baseline (January 2021). Researchers were able to identify participants during data collection, but all data were anonymized by code before data entry and statistical analysis.

**Table 1. Study participants (N = 89).**

| Schools | 4GTs | HPETs | 4th-grade classes students |
|---------|------|-------|---------------------------|
| #1 | 1 | 1 | 25 *(12 boys & 13 girls)* |
| #2 | 1 | 1 | 17 *(7 boys & 10 girls)* |
| #3 | 2 | 1 | 18 *(8 boys & 10 girls)* |
| | | | 22 *(9 boys & 13 girls)* |

4GTs: Fourth-grade teachers; HPETs: health and physical education teachers; 4th-grade classes students: children aged 9–11 years old.

### Instruments to assess the implantation of the PAL program

Qualitative data was collected using various methods, including teacher experimental notes, group interviews with teachers, and focus groups (FGs) with students. The 4GTs completed an observation grid for each PAL lesson to collect students' observance and subjective evaluation of the lesson. The 4GTs identified inadequate elements of the program and proposed solutions to address them, while technical bugs (e.g., visual bugs) were reported to improve future iterations of the PAL program.

After the 8-week intervention period, semi-structured interviews were conducted with teachers using a guide covering major thematic questions present in Table 2 with additional questions as needed [27]. The interviews were conducted in three sessions, with two or three participants in each session, and were 60 to 90 minutes long: interview #1: n = 2 [4GT #1 & 4GT #2]; interview #2: n = 2 [4GT #3 & 4GT #4]; and interview #3: n = 3 [HPET #1, HPET #2 & HPET #3]. The interviews were structured as informal conversations and conducted via Zoom Meeting and recorded for verbatim transcription due to the COVID-19 pandemic restrictions. A confidentiality statement was provided at the beginning of each interview to ensure the privacy of the participants [28].

The four FGs with students were conducted in their respective classrooms and lasted between 35 to 50 minutes each: FG #1, n = 25; FG #2, n = 17; FG #3, n = 18 & FG #4, n = 22 (See Table 1 for more details). Due to the COVID-19 pandemic, it was not feasible to form smaller, random groups of students, as recommended by Morra-Imas, Morra [28]. The discussion was stimulated using themes and sub-themes presented in Table 3, which were based on the research questions. The moderator (first author) facilitated the discussions, allowing students to expand upon their responses. The structure of the FG discussion followed the guidelines outlined by Morra-Imas, Morra [28], as shown in Table 4. Students were encouraged to participate by raising their hands. The teacher ensured that the students respected each other's turn to speak and encouraged quieter students to share their thoughts. To facilitate transcription of the verbatim, an Owl video camera and microphone (Owl Labs, US) were used, and a researcher assistant was present to ensure is proper functioning.

### Procedures

**Co-design framework.** The co-design of the PAL program was carried out from October 2019 to October 2020 using a multidisciplinary framework that resulted in constructivist

**Table 2. The semi-directed interviews guide for teachers.**

| Themes and sub-themes | Description |
| --- | --- |
| Theme 1 | PAL program |
| Sub-theme 1.1 | Description and purpose of the PAL program |
| Theme 2 | Action research process |
| Sub-theme 2.1 | Personal and professional benefits |
| Sub-theme 2.2 | Challenges |
| Theme 3 | PAL program's assessment |
| Sub-theme 3.1 | Overall outcomes |
| Sub-theme 3.2 | Students and classroom behaviour outcomes |
| Sub-theme 3.3 | Learning and academic outcomes |
| Theme 4 | Planning for future iterations |
| Sub-theme 4.1 | Key success factors to PAL implantation |
| Sub-theme 4.2 | Perceived barriers to PAL implantation |
| Sub-theme 4.3 | Future utilization and suggested modalities |

**Table 3. The FGs discussion guide for students.**

| Themes and sub-themes | Description |
|---|---|
| Theme 1 | General PAL program's assessment |
| Sub-theme 1.1 | Program's enjoying aspects to keep |
| Sub-theme 1.2 | Program's uninteresting aspects to be removed |
| Theme 2 | Specific PAL program assessment |
| Sub-theme 2.1 | Overall outcomes |
| Sub-theme 2.2 | Physical/motor outcomes |
| Sub-theme 2.3 | Learning and academic outcomes |
| Theme 3 | Future iterations |
| Sub-theme 3.1 | Suggested changes (add and remove) |

learning [29]. The co-design process was based on the Double-Diamond model [30], which is an open innovation framework consisting of four phases: Discover, Define, Develop (co-design), and Deliver (implementation). Fig 1 provides a visual representation of the model.

During the first phase (Discover), the research team conducted a scientific literature review to identify an innovative solution to increase PA levels among school-aged children while also improving the development of children with neurodevelopmental disorders, such as attention deficit disorder with or without hyperactivity [31], which is a prevalent condition in the study's region [32]. The second and third phases (Define & Develop) involved the collaboration of school practitioners, who joined the research team to select from the identified possibilities the PAL program property that would be suitable for the school environment and the teaching curriculum. Three meetings were scheduled, during which school practitioners discussed various topics that were essential to the design process. Table 5 summarizes the topics covered during each meeting. After these meeting, the school practitioners continued to work on a collaborative tool that guided designers in producing different iteration of the PAL program. The final phase (Deliver) involved the implementation of the PAL program in classrooms to evaluate the prototype.

**Implementation framework.** The implementation of the PAL program consisted of 24 sessions delivered for 8 weeks, with three sessions per week. A crossover design was used,

**Table 4. The Structure of FGs.**

| Phase | Actions |
|---|---|
| I. Foreword | Moderator puts participants at ease by beginning with ice-breaking questions |
| | Moderator explains the purpose of the focus group |
| | Moderator provides ground rules |
| II. Introduction | Moderator stimulates group interaction and thinking about the topic |
| | Moderator stats with the least threatening and simplest questions |
| III. Main body of group discussion | Moderator moves to more threatening or sensitive and complex questions |
| | Moderator elicits deep responses |
| | Broad participation is ensured |
| IV. Closure | Moderator ends with closure-type questions |
| | Moderator summarizes and refines key themes |
| | Moderator invites participants to provide final comments, plus anything missed or anything else participants would like the evaluation team to know |
| | Moderator thanks participants |

Adapted from Morra-Imas, Morra [28]

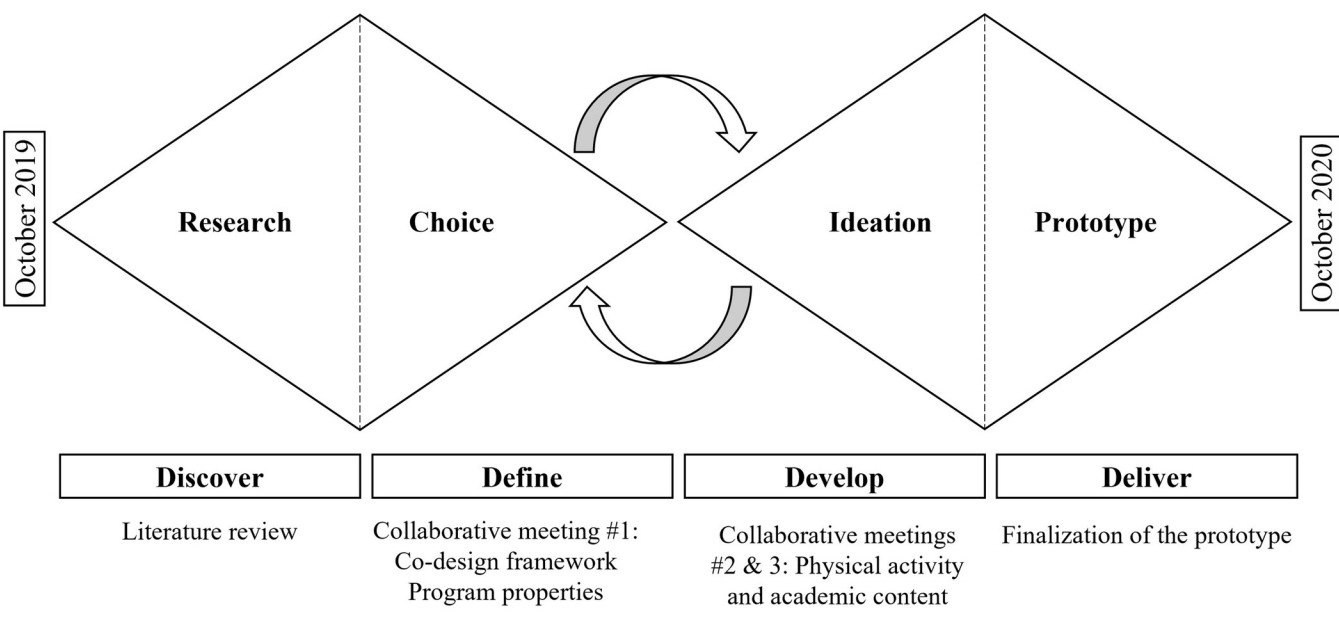

**Fig 1. Co-design timeline.** Adapted from the Double Diamond design process model, developed by the British Design Council.

where the PAL program was initially implemented in two classrooms during January and February 2021, followed by implementation in the remaining two classrooms during March and April 2021 (see Fig 2). To gather feedback and insights, teachers' interviews and FGs with students were conducted in the week following the completion of the PAL implementation in their respective classrooms.

## Data handling and analysis

The data collected from the interviews and FGs were transcribed verbatim to ensure accuracy [33]. An inductive approach to thematic analysis was used to analyze the data collected from both students and teachers [34]. This approach allowed for the identification of patterns and themes that emerged from the data, rather than imposing pre-existing categories on the data.

Qualitative data were coded and categorized using NVivo 12 software (QSR International, US). Students who did not provide consent for the study were assigned the pseudonym 'Student X' in the verbatim, and their quotes were excluded from the analysis data. This inter-rater agreement process was followed by the main author and a research assistant to ensure consistency in the analysis. The resulting thematic tree map was tested multiple times and validated by the research team (inter-rater validation operations), leading to the identification of main

**Table 5. Co-design planification.**

| Meeting | Themes |
|---------|--------|
| 1 | Study contextualization |
| | Co-design framework |
| | PAL's sessions properties |
| 2 | Academic themes |
| | Interactive elements |
| 3 | Academic content by theme |
| | PA & movements content |

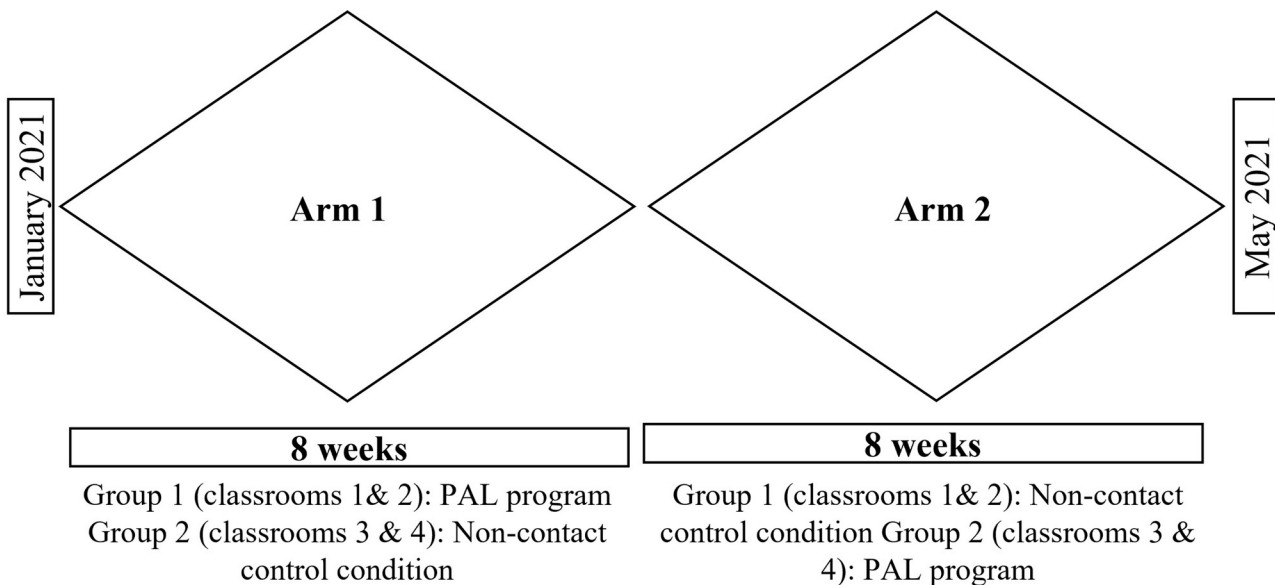

**Fig 2. Implementation timeline.** The crossover design from January 2021 to May 2021.

themes [34]. Notably, all quotes presented in this article were translated from French to English by the researchers, with attention given to maintaining equivalence of meaning [35].

## Results

This section presents the findings of the thematic analysis conducted on the data generated by the semi-structured interviews and the FGs. The main themes were related to the objectives of the study which were 1) to co-design and 2) to implement a PAL program in elementary schools.

### Objective #1: Co-designing a PAL program

To design an effective PAL program, the involvement of teachers has been recognized as a critical element. In this regard, an in-depth analysis of verbatim data was conducted to gain valuable insights into their perspectives and experiences regarding the co-design process. Through this analysis, emergent themes surfaced in the co-design process such as their level of involvement, the outcome *[the PAL program]*, and the emotions they experienced during the co-design process. These themes not only provide a comprehensive understanding of the teacher's role in designing the PAL program but also facilitate the development of strategies to ensure their active participation and engagement in future initiatives.

**Teachers' involvement in the co-design process.**   During the post-experimental interviews, all teachers involved were asked why they chose to become engaged at that point. The interview data showed that some teachers participated in the action research project for their students. HPETs highlighted ″the opportunity to increase students' PA behaviour while learning through movement″. Other teachers also mentioned ″being grateful for the opportunity to participate in the co-design of a useful pedagogical tool, and to be aware of the outcomes afterwards″.

> ″. . .it was a motivating project for the students, it was nice to tell them that we were thefortunate involved in this project. . . I was happy to be part of the research project because this tool could be reapplied and integrated into teachers' educational practice.″ 4GT #2

During the co-design process, the teachers were involved in the key elements of the PAL program, such as the academic content, PA integration, and artistic creation of the avatar. The 4GTs worked mainly on academic content. After agreeing on academic themes (see Table 6), they worked separately to develop the framework for each thematic activity to be incorporated into the PAL program. All 4GTs described that part of the co-design process as "a huge work-load". The specific tasks of the HPETs were to identify the physical movements of the PAL program (see Table 7). The fourteen movements were chosen 1) to be fun, 2) doable behind the desk, and 3) to increase heart rate. The intensity of the movements (low, moderate or vigorous) was classified based on the Youth Compendium of Physical Activities [36]. For the avatar, six sketches were first presented to the teachers. These sketches, designed by a digital artist, were based on three criteria: being 1) empathic; 2) non-gendered; and 3) intended for an audience of Canadian children aged 9–11. Teachers overwhelmingly voted for a raccoon, which was then named Ratou (see Fig 3).

**Overview of the PAL program–Cogni-Actif.**   The active involvement of teachers in the co-design process led to the development of a unique PAL program called Cogni-Actif. According to the teachers, Cogni-Actif is an ″easy-to-use and fun way to promote learning through movement″ that "combines physical and academic activities into 15-minute video clips″. It ″helps to review academic material and consolidate learning by incorporating move-ment into the classroom″. They highlighted the program as a ″motivator for the children″ and ″a good way to move in the classroom″ that ″helps with concentration and learning″. Finally, the teachers noted that Cogni-Actif is ″an end-to-end solution that does not require any addi-tional preparation from them″. This feature eliminates the need for teachers to spend extra time creating activities, enabling them to focus on delivering high-quality instructions.

**Emotions felt by teachers during the co-design process.**   The co-design process resulted in the emergence of two distinct emotional states: ″stress and pride″. The perceived stress dur-ing the process was mainly attributed to the pandemic context, while the feeling of pride was

**Table 6. Academic lessons themes.**

| Lesson # - Themes | Lesson # - Themes |
|---|---|
| 1 - Mathematic Table 2 | 17 - Mathematic Table 9 |
| 2 - Gender of French nouns | 18 - Simple future (-Er verbs) |
| 3 - Mathematic Table 3 | 19 - Mathematic Table 10 |
| 4 - Plural of French nouns | 20 - Simple future (-Ir verbs) |
| 5 - Mathematic Table 4 | 21 - Comparison of numbers |
| 6 - Words of the same family | 22 - Imperfect indicative (-Er verbs) |
| 7 - Mathematic Table 5 | 23 - Solids |
| 8 - Categories of word classes | 24 - Imperfect indicative (-Ir verbs) |
| 9 - Mathematic Table 2 to 5 | 25* - Angles |
| 10 - French noun group | 26* - Conditional present (-Er verbs) |
| 11 - Mathematic Table 6 | 27* - Quadrilaterals |
| 12 - Verbs conjugation (infinitive) | 28* - Conditional present (-Ir verbs) |
| 13 - Mathematic Table 7 | 29* - Polygons |
| 14 - Present indicative of (-Er verbs) | 30* - All tenses (-Er verbs) |
| 15 - Mathematic Table 8 | 31* - Fraction readings |
| 16 - Present indicative (Ir verbs) | 32* - Units of measurement |

*Academic lessons not tested (backup due to COVID-19 interruption)

**Table 7. Physical movements of the PAL program.**

| Movements | Intensity |
|---|---|
| One-Foot Balance Standing | Low |
| Walking | Low |
| Boxing | Moderate |
| Drum solo | Moderate |
| Hula-Hoop Dancing | Moderate |
| Squat | Moderate |
| Cross-Country Skiing | Vigorous |
| Cross Jumping | Vigorous |
| Frog Jumping | Vigorous |
| Heel to Butt Jumping | Vigorous |
| High Knee Running | Vigorous |
| Running | Vigorous |
| Scissors Jumping | Vigorous |
| Skipping | Vigorous |

strongly associated with the active involvement of teachers and students in the co-design process.

> "*Yes, it was stressful [COVID-91 period], but while we were working on the co-design of the PAL program, we weren't thinking about the pandemic, it kept us busy. . .*" 4GT #1

> "*. . .it required more work than I initially expected when the project have been presented to us. . . honestly, I didn't expect that, but am I'm proud I did it [the co-design]. . . Every time I had visitors in my classroom, I showed them the program, and they were all very surprised that we were doing this in my classroom. They all told me that they wished their children could have experienced it too, to motivate them to go to school. I really think it's a great program and our students are lucky to experience it, I would definitely do it again. . . When they [the students] talk about their experience, it's always with great joy. The pride I felt, they felt it too. . .*". 4GT #1

Despite the challenges posed by the pandemic context, the co-design process involving teachers played a crucial role in developing a PAL program that effectively integrates PA and academic content for elementary school students. The successful collaboration between educators and researchers demonstrated by the co-design process offers a valuable tool for promoting PA and academic learning in elementary school settings.

## Objective #2: Implementing the PAL program in elementary schools

The second objective of the study was to assess the implementation of the PAL program in school classes. The inductive analysis of the PAL program implementation highlighted four main components: 1) Implementation context, 2) Perceived Outcomes, 3) Assessment of the PAL program components, and 4) potential future use of the PAL program.

**The implementation context.** Despite the unprecedented challenges posed by the COVID-19 pandemic, the study continued as it was deemed a priority by the school practitioners involved and their managers. Even if the pandemic was not a planned theme of interviews or FGs, teachers and students repeatedly reported how it affected the study's processes, and that the pandemic context likely influenced the study's outcomes. For instance, students

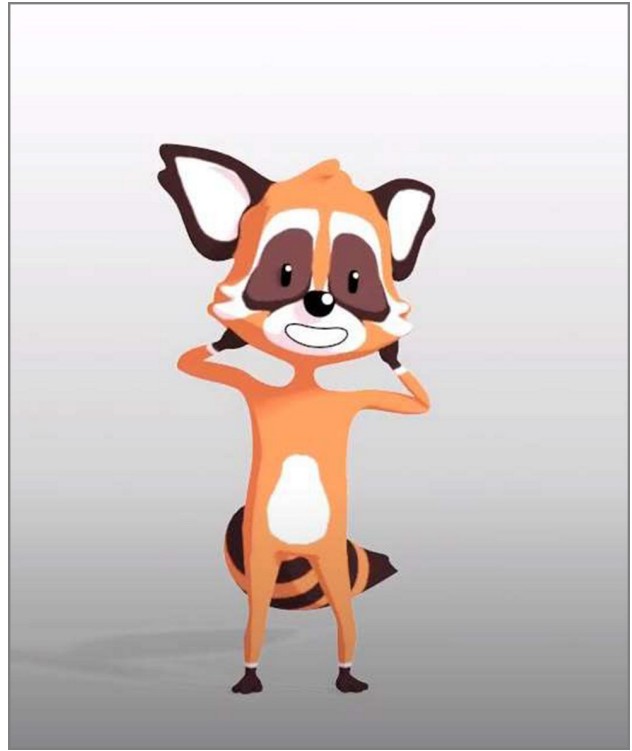

**Fig 3. PAL program's avatar named Ratou.** Illustrated by Matoub, I. 2019.

who experienced the PAL program from March to May 2021 (n = 40) had to were masks in class, which might have affected their willingness to complete the sessions with the prescribed intensity. In addition, one classroom switched to remote learning during the PAL implementation. These students completed five home lessons supervised by their teacher via computer video.

> "*Moving to remote learning was a challenge. . . It was still going well [implementing the PAL program], I was working with two computers at home, one to share the PAL program with the students and one to watch them perform. . . I couldn't watch them all, but it was doable.*"

> \* To keep the 4GT name's anonymous, this quote is not associate to the participant code.

Despite challenges caused by the COVID-19 pandemic, the study continued, providing valuable insight into the feasibility of implementing a PAL program during a pandemic and its positive outcomes.

**Perceived outcomes by the teachers and the students.** This section specifically delves into the perceived outcomes of the PAL program implementation as reported by teachers and students. Participants reported several benefits related to determinants associated with students' learning, attention and focusing ability, PA levels, and quality of sleep. Examining these outcomes can provide insights into the effectiveness of the PAL program and its potential impact on elementary school students.

*Learning support.* According to the teachers, the PAL program has the ″potential to improve students' learning″ as they observed, ″students learned faster and retrained more easily, particularly for math concepts.″ They confirmed that ″combining movement and academic

effectively consolidates learning and anchors knowledge, particularly for concepts that are easy to learn through practice, such as multiplication tables." The teachers suggested that combined learning, which involved movement, "benefits students in many ways and is effective because it engages them." Students also shared this impression during the FGs. Lastly, teachers reported that the implementation of the PAL program in class was a "good way to review concepts quickly seen in the previous year due to COVID-19, which allowed for the consolidation and deepening of knowledge."

> "*I think that when you're learning and doing a gesture or movement at the same time, it gives an additional anchor.*" *4GT #1*

> "*. . .for the kids, as long as the learning is through play, it's clear that the interest is greater because they don't feel like they're learning. . .*" *4GT #3*

> "*. . . it makes it easier for me to retain information because I'm having fun at the same time.*" *Student #51*

> "*I noticed that it makes us learn more quickly. And I also think that math got into my head faster.*"*Student #28*

*Improved attention and focus.* The PAL program improves students' attention, focus, and on-task behaviour, which in turn contributed to their learning. The 4GTs reported that the "classroom was quieter after the program" and that "students were more willing to learn at that time. " They also mentioned "teaching more difficult concepts immediately after the program to take advantage of students' increased attention."

> "*I definitely took advantage of it to do a teaching period [immediately after the PAL program] because I had their attention. . .. That effect faded over the course of the day, as the day went on, it went back to normal.*" *4GT #1*

> "*. . .after Ratou [the program character], I always took the opportunity to teach more complex material (e.g., fractions in math) . . . they were more inclined because they just moved.*" *4GT #4*

> "*It was a good start to my day. Afterwards, I was ready to work, I had more focus, and the class was quieter.*" *Student #38*

*Improved PA levels.* The PAL program was found to be an effective way to increase students' PA levels by adding movements during class time. Students reported "enjoying being active in class," and surprisingly, this enthusiasm translated to reported, "increased PA outside the classroom and a reduction in screen time at home."

> "*What I liked best about the program is that we are physically active . . . it's important for our body to be physically active.*"*student #48*

> "*. . . at home I do more physical activity, I'm more motivated and I go outside more often. . . I've also noticed at home that I'm less connected to my computer or my tablet.*" *Student #45*

> " *. . . before the Cogni-Actif project, I wasn't going outside to play with my friends much, but now I go a lot more, I go almost every day that we go to school. *" *Student #22*

> "*Ratou gave me idea how to train. . . it helped me and my sister to respect the instructions of screen time . . . we went a little more outside. Usually, we were often on our phones, then we turned them of a little more . . .*" *Student #32*

Despite the increase in PA due to the PAL program, neither the HEPTs nor the 4GTs observed any changes in the children's physical fitness. However, several students noticed ″improvement in their endurance and overall fitness.″

"*I believe that my endurance has improved.*" *Student #43*

"*The first week we started Cogni-Actif program, I had a lot of pain in my legs. But then, after the first week, it was okay, and by the end, I didn't feel it anymore.*" *Student #45*

"*I feel that I'm running a little faster.*" *Student #46*

"*I'm more physically fit.*" *Student #24*

*Improved quality of sleep.* Most students reported ″experiencing better sleep quality and an earlier onset of sleep on days when day participated in the PAL program.″ They attributed this to the fact that they were ″more physically active and tired on those days.″ In fact, one parent even contacted the 4GT to share that her daughter was going to bed earlier on days when the Cogni-Actif program was in effect. These observations suggest that while physical fitness improvements may not be immediately apparent, increased PA can have positive effects on sleep quality and duration.

**Assessment of the PAL program components.** The PAL program's assessment by students and teachers identified three main components: Academic Content, Physical Movements, Graphics and Sound. For each component, they provided key solutions to improve the program to make it more enjoyable and relevant to the school setting.

*Academic content.* According to the 4GTs, the curriculum they selected for the PAL program during the co-design process was ″mostly relevant and on target for grade 4.″ However, they agreed that ″Maths material was more relevant than French ones.″ Students also mentioned that they preferred Math to French lessons while using the PAL program. Only conjugation seemed appropriate for reviewing in a PAL program from the French themes listed in Table 6. The ″academic content's difficulty level was appropriate for most students,″ but some aspects need to be adjusted for the next iteration. According to the 4GTs, the PAL program was ″too difficult for students with special needs (e.g., dysphasia and dyslexia).″ The students addressed the fact that they ″did not have enough time to find the answer″ before it appeared on the interactive board on some occasions. Finally, 4GTs proposed ″covering fewer academic topics but adding lessons to increase repetition.″ They stated that more repetition would anchor the knowledge more firmly and make it a better review tool.

"*I would remove some themes and repeat the simpler themes like addition, multiplication, and conjugation . . . I would use the Ratou session in class once a day as a memorization tool and spend the week on the same theme. . .*"*4GT #2*

*Physical movements.* Physical movements are a huge part of the PAL program. When asked which movements they preferred, the students named most of them. Generally, the ″intensity was usually adequate except for a few occasions when the intensity was too high for the duration of the exercise.″ The unanimous ones were the ″boxing and the drum solo, which are more playful movements.″ A lack of variety emerged from discussions with students and was confirmed by the teachers. It was the walking movement that created the most dissatisfaction because it came back every session to allow students to calm down between moderate or vigorous exercise. It was suggested that ″new, lower-intensity movements be identified to replace the walking.″ The students also suggested ″adding some new ones, such as hip-hop moves or

basketball shots." In addition, all 4GTs mentioned that when Ratou's pace was unrealistic (too fast or too slow), it distracted students from the intention of reviewing French or Maths lessons. As a key solution, the "animations must be redesigned for some movements."

*Graphic and sound elements.* The graphic and sound elements of the program were designed to stimulate student motivation and participation. The students unanimously liked the character Ratou. According to the teachers, the program was "visually attractive but would need something more stimulating to maintain students' interest" because children are used to colourful and animated video games. The students were impressed at the beginning of the program, but this faded as the program went on. Indeed, in the FGs, the students named several solutions to "make the PAL program more stimulating" particularly to "add graphic variety" to the PAL program.

"*Ratou is a prototype that could evolved in a video game.*" 4GT #2

"*. . .at the beginning Ratou was wow, wow, wow, but at some point Ratou had little flaws, ah it's the same, the colors were the same, Ratou is always dressed the same . . .*" 4GT#4

On another hand, the "music might be more relevant to a youthful audience" and less repetitive. Most students said they would prefer music they know and also to be able to select the songs. As mentioned earlier, students would add "narration and sound effects."

"*It could be like in Mario Bros a* "DING" *sound at the end. . .*" student #50

**Potential uses of the PAL program.** The finding of this study indicates that the PAL program has the potential for use in elementary school classrooms. The 4GTs expressed interest in using the PAL program in their classroom in the future. According to them, the PAL program "should be used in the classroom over a long period of time to maximize the benefits, from October to April." The 4GTs suggested avoiding using the program during September, May, and June due to weather conditions in Quebec. The program should be "used daily", reviewing one topic per week. The 4GTs also provided suggestions for optimizing its use. The suggestions included adapting the PAL program into a "video game", incorporating an "individual at home component" to the game, and "adapting the PAL program for other grade levels."

*Adapt the PAL program into a video game.* The interaction was identified as a crucial component of the PAL program by both the 4GTs and the students. As a result, they suggested adapting the program into a video game that would allow for "increased interactivity." The video game version would allow students to "interact with the character and solve quests." Students suggested incorporating a collection feature, which would add a motivating aspect to the game. "Messages of encouragement" from the character would also be beneficial to students. The video game version should give the teachers the ability to "select the difficulty level and activities".

"*I see Ratou as a prototype of something that evolved . . . the students are used to video games. . . and it would allow the teacher to select the activity . . . to students to see their progress . . . to become a kind of quest.*" 4GT #2

"*They [students] wanted to collect something. . . . it could be an educational game with pedagogical capsule and something to collect by students.*" 4GT#1

"*Ratou could say messages of encouragement as* "You did the first part, Bravo" *or* "That's good, keep going, you're well on your way." *Student #18*

*Add an individual component.* The 4GTs suggested adding an ″individual at home component″ to the PAL program in addition to the group sessions conducted in class. This would allow students to ″review the material at their own pace″ at home via an accessible platform. The teacher would be able to ″monitor student progress″ and evaluate the level of each student.

*Adapt the program for other grade levels.* The PAL program was designed for students in the 2$^{nd}$ cycle, specifically in fourth grade (9–10 years old). However, teachers suggested ″adapting it for other grade levels, including 1$^{st}$ cycle (1$^{st}$ and 2$^{nd}$ grades) for addition and subtraction review, and 3$^{rd}$ cycle (5$^{th}$ and 6$^{th}$ grades) for English second language concepts.″ To avoid redundancy, different ″characters could be created for each cycle″; the concept is easily adaptable for each elementary school levels by ″following elementary curriculum.″

In conclusion, the PAL program has the potential to be an effective tool for PA promotion in elementary school classrooms. The suggestions made by the 4GTs for optimizing the PAL program should be considered for the next iterative co-design process. All teachers involved agreed to be part of the next iteration co-design process of the PAL program. However, they suggested recruiting a larger team to facilitate the co-design process, which hopefully will not occur in a pandemic period.

## Discussion

The integration of PA with academic learning is an area of growing interest in education, as schools face the challenge of providing enough opportunities for children to engage in regular PA while also meeting academic objectives [17, 37, 38]. The present study aimed to address this challenge by developing and evaluating the implementation of a PAL program in school classrooms over 8 weeks. Thematic analysis of interviews with 7 teachers and FGs with 82 students provided valuable insights into their perceptions of the PAL program. While the study's findings have significant practical implications for the development and deployment of future PAL programs, it is important to exercise caution in their application due to some limitations of the study, which will be mentioned later in the discussion. Future research should address these limitations to enhance the generalizability of the findings and improve the effectiveness of future PAL programs.

### Practical implications

This study provides valuable insights for school practitioners and researchers looking to implement PAL programs. The finding highlights the benefits of integrating PA with academics in the classroom, which can help schools meet both PA and educational objectives. Additionally, the study emphasizes the importance of involving practitioners in the co-design process of PAL programs, which can facilitate the development of effective and tailored solutions that fit within the school setting and address specific needs and challenges. Finally, the study suggests that technology can play a crucial role in supporting PA practice at school, offering promising avenues for future research and program development.

**Benefits of integrating the PAL program into the classroom.** The World Health Organization (WHO) recommends that children engage in at least 60 minutes of moderate- to vigorous-intensity PA each day, and one way to help achieve this goal is by implementing physical activity during school hours while still achieving academic objectives [1]. This approach is reflected in government policies around the world. For example, in Quebec, where the study was conducted, the government has implemented the ″*À l'école on Bouge*! ″ policy, which requires schools to provide at least 60 minutes of PA per day [39].

The study confirms that integrating a PAL program can facilitate meeting the daily PA practice recommendations, not only by adding 10–12 minutes of moderate- to vigorous-

intensity PA in the classroom but also by leading to an increase of PA practice outside of the classroom and a decrease in screen time, which is in line with the World Health Organization 24-hours guidelines for children's health [1]. The integration of learning with movement was found to be an effective way to review academic content and improve student attention and focus, leading to a better disposition for the student to learn.

However, it is important to note that a quantitative measure of PA levels by accelerometer measure would be necessary to better understand the impact of PAL programs in school on the overall PA practice of children [40, 41]. Despite this, the findings of this study are consistent with previous literature on the benefits of integrating PA with academic learning, enhancing attention and focus, which benefits teachers' pedagogy in the hours following the PAL program [7, 38, 42]. These outcomes can be motivating factors for teachers to implement PAL programs in their classrooms, providing health and academic benefits as well as promoting PA among their students [7, 42].

**Co-designing solutions with practitioners.** Co-designing solution with end-users lead to a more holistic view of the problem and a better understanding of their needs resulting in tailored solutions [43]. The study findings support this approach, as co-designing the PAL program with teachers and other school staff facilitated the development of a tool that fits within the school setting and is related to student targets. Additionally, the involvement of teachers in the co-design process facilitated the program's implementation in the school. This approach contributed to a sense of pride in what was accomplished, which is a success factor in the implementation and sustainability of the solution. This finding aligns with previous research on co-design, which has shown that involving end-users in the design process is critical to ensuring that the program fits within the school setting and addresses the specific needs and challenges of the school community [7].

**The potential of technology to support PA practice.** The potential of technology to support children's PA practice has been recognized by several studies [44–46], and this study's findings further highlight its usefulness. Specifically, the use of multimedia video technology to diffuse the PAL program in the classroom facilitated its implementation by providing an easy-to-use solution for teachers.

Interestingly, both teachers and students involved in the study suggested enhancing the use of digital technology by developing an exergame to implement the PAL program. This aligns with the public policies to increase the use of digital technologies in schools [47]. Developing serious games that support both PA practice and learning objectives can be a promising way to further enhance the effectiveness of PAL programs.

## Limitations

Limitations should be considered when interpreting the results of this study. First, the fact that the PAL program was implemented only in the classrooms of the 4 GTs who were involved in the co-design process may have influenced their perceptions of the program's benefits, and therefore limit the generalizability of the findings. Second, the size of the focus groups used for data collection among the students exceeded the recommended size, which may have generated fear of speaking in front of others and potentially affected their perceptions [28]. However, data saturation was achieved and the creation of an atmosphere of trust and openness during the focus groups by the moderator may have helped mitigate this limitation. It is important to consider these limitations when applying the findings to other contexts or populations.

### Future research

Future research could further explore the generalizability of our findings by conducting similar studies in different settings or with different populations. In addition, it would be valuable to evaluate the implementation of PAL programs with teachers who did not participate in the co-design process, to determine if similar benefits can be achieved. Reliable quantitative measures should be obtained to confirm the results obtained, particularly with regard to the increase in PA participation, the decrease in screen time outside of school, and the effects on classroom behaviour and academic performance. These measures could help to better understand the impact of PAL programs on children's overall health and well-being and inform the development of more effective interventions in the future.

Furthermore, future research could consider the development of a new iteration of the PAL program that considers the feedback provided by teachers and students. This could involve the development of a serious game that allows for more variety and interaction to maintain a high level of engagement among students. The game could also allow teachers to select the most appropriate difficulty level for their students based on their specific needs. By addressing these areas of improvement, we can continue to improve the effectiveness of PAL programs and enhance children's physical activity levels and overall health.

## Conclusion

In conclusion, this study highlights the potential of integrating physical activity with academic learning and the importance of co-designing programs with practitioners to maximize their effectiveness. The findings suggest that technology can be a powerful tool to support PAL practices in promoting both physical activity and academic achievement among school-aged children. The study also underscores the need for further research to explore the broader societal implications of PAL programs, including their potential to contribute to public health goals and reduce health disparities among different groups of students. Ultimately, the development of effective PAL programs has the potential to improve the health and well-being of children and promote positive academic outcomes.

## Supporting information

**S1 Checklist. PLOS ONE clinical studies checklist.**
(DOCX)

## Acknowledgments

The authors thank Anne-Julie Bouchard, Audrey Fortin, Camille Fournier, Francis Maltais and Sabrina Pilote for their valuable contributions to this study. We are also grateful to the Lac-Saint-Jean school service centre (Québec) and the administration and staff involved in this study as well as all the research assistants.

## Author Contributions

**Conceptualization:** Laurie Simard, Julie Bouchard, Martin Lavallière, Tommy Chevrette.

**Data curation:** Laurie Simard.

**Formal analysis:** Laurie Simard, Julie Bouchard, Martin Lavallière, Tommy Chevrette.

**Funding acquisition:** Laurie Simard, Tommy Chevrette.

**Investigation:** Laurie Simard.

**Methodology:** Laurie Simard, Julie Bouchard, Martin Lavallière, Tommy Chevrette.

**Project administration:** Laurie Simard, Martin Lavallière, Tommy Chevrette.

**Supervision:** Martin Lavallière, Tommy Chevrette.

**Validation:** Julie Bouchard.

**Writing – original draft:** Laurie Simard, Martin Lavallière, Tommy Chevrette.

**Writing – review & editing:** Laurie Simard, Julie Bouchard, Martin Lavallière, Tommy Chevrette.

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
