## [Decision Letter · Decision Letter 0]

6 Jun 2023

PONE-D-23-12396Promoting Physical Activity and Academic Achievement through Physically Active Learning: Qualitative Perspectives of Co-Design and Implementation ProcessesPLOS ONE

Dear Dr. Simard,

Thank you for submitting your manuscript to PLOS ONE. After careful consideration, we feel that it has merit but does not fully meet PLOS ONE’s publication criteria as it currently stands. Therefore, we invite you to submit a revised version of the manuscript that addresses the points raised during the review process.

The feedback provided by the reviewers highlights several areas that require attention in order to improve the study. Specifically, they have emphasized the need for a better-articulated objective to support the rationale of the study. Additionally, the reviewers have requested more background information in the introduction and have pointed out that the methods section is currently confusing. They have also noted the necessity for additional references to support the arguments proposed by the authors. Please refer to the reviewers' comments for the details. Please note that Reviewer 2 has embedded his/her comments as sticky notes within a PDF file.

Apart from the reviewers' comments, the editor has raised two additional points that need to be addressed:

1. On page 11, the editor requests clarification on the implementation framework used to guide the implementation process. The current description provided by the authors focuses on the study design and timeline for implementing and evaluating the PAL program. Therefore, it is necessary to explicitly state the specific implementation framework that was utilized.

2. Please provide a comprehensive description of the implementation process of the PAL program. The authors should ensure that all the steps and components involved in implementing the program are explicitly explained, providing a clear understanding of how it was executed.

We look forward to receiving your revised manuscript.

Kind regards,

Erica Y. Lau

Academic Editor

PLOS ONE

 [copy in funding statement]. 

“Cogni-Actif, the PAL program, has been co-designed in an action research project led by the authors. No other potential conflict of interest was reported by the authors.”

Please respond by return email with your amended Competing Interests Statement and we will change the online submission form on your behalf.

5. We note that Figure 2 in your submission contain copyrighted images. All PLOS content is published under the Creative Commons Attribution License (CC BY 4.0), which means that the manuscript, images, and Supporting Information files will be freely available online, and any third party is permitted to access, download, copy, distribute, and use these materials in any way, even commercially, with proper attribution. For more information, see our copyright guidelines: http://journals.plos.org/plosone/s/licenses-and-copyright.

b.If you are unable to obtain permission from the original copyright holder to publish these figures under the CC BY 4.0 license or if the copyright holder’s requirements are incompatible with the CC BY 4.0 license, please either i) remove the figure or ii) supply a replacement figure that complies with the CC BY 4.0 license. Please check copyright information on all replacement figures and update the figure caption with source information. If applicable, please specify in the figure caption text when a figure is similar but not identical to the original image and is therefore for illustrative purposes only.

Reviewers' comments:

Reviewer's Responses to Questions

**Comments to the Author**

1. Is the manuscript technically sound, and do the data support the conclusions?

Reviewer #1: Partly

Reviewer #2: Yes

2. Has the statistical analysis been performed appropriately and rigorously? 

Reviewer #1: N/A

Reviewer #2: N/A

3. Have the authors made all data underlying the findings in their manuscript fully available?

Reviewer #1: Yes

Reviewer #2: Yes

4. Is the manuscript presented in an intelligible fashion and written in standard English?

Reviewer #1: Yes

Reviewer #2: Yes

5. Review Comments to the Author

Reviewer #1: The manuscript entitled: Promoting physical activity and academic achievement through physically active learning: qualitative perspectives of co-design and implementation processes describes teachers’ and students’ perspectives on the processes related to developing and implementing a curriculum designed to enhance movement and learning. The findings of this study will be useful and informative to the PLOS ONE audience however more background information is needed in the introduction and the methods as currently written is confusing.

Reviewer #2: First of all, thanks again for offering the opportunity to act as a referee for PLOS ONE journal .

The investigation was properly developed and the paper was clear. I am just attaching a pdf with 8 comments to solve.

Thanks again.

6. PLOS authors have the option to publish the peer review history of their article (what does this mean?). If published, this will include your full peer review and any attached files.

Reviewer #1: No

Reviewer #2: **Yes: **Carlos Capella-Peris

---

## [Author Response · Author response to Decision Letter 0]

21 Sep 2023

Dear reviewers,

We would like to express our sincere gratitude for the time you dedicated to evaluating our manuscript and for the quality of your constructive feedback. Your input has been invaluable in improving our work.

We are pleased to submit a revised version of our manuscript in response to your comments. We have addressed each point you raised and made appropriate modifications. You will find a detailed account of our responses in the document titled "Response to Reviewers," which we have attached to this submission.

Once again, thank you for your commitment to enhancing our research. We look forward to receiving any additional feedback.

Best regards,

---

## [Decision Letter · Decision Letter 1]

2 Nov 2023

Promoting Physical Activity and Academic Achievement through Physically Active Learning: Qualitative Perspectives of Co-Design and Implementation Processes

PONE-D-23-12396R1

Dear Dr. Simard,

We’re pleased to inform you that your manuscript has been judged scientifically suitable for publication and will be formally accepted for publication once it meets all outstanding technical requirements.

Kind regards,

Josephine N. Booth

Academic Editor

PLOS ONE

Additional Editor Comments (optional):

I think you have done well to address the reviewers concerns. Please ensure you do a thorough check for typos and spelling errors in the manuscript though.

Reviewers' comments:

Reviewer's Responses to Questions

**Comments to the Author**

1. If the authors have adequately addressed your comments raised in a previous round of review and you feel that this manuscript is now acceptable for publication, you may indicate that here to bypass the “Comments to the Author” section, enter your conflict of interest statement in the “Confidential to Editor” section, and submit your "Accept" recommendation.

Reviewer #2: All comments have been addressed

2. Is the manuscript technically sound, and do the data support the conclusions?

Reviewer #2: Yes

3. Has the statistical analysis been performed appropriately and rigorously? 

Reviewer #2: N/A

4. Have the authors made all data underlying the findings in their manuscript fully available?

Reviewer #2: Yes

5. Is the manuscript presented in an intelligible fashion and written in standard English?

Reviewer #2: Yes

6. Review Comments to the Author

Reviewer #2: All comments have been addressed. Thanks for your consideration and for your carefully revision of the manuscript.

7. PLOS authors have the option to publish the peer review history of their article (what does this mean?). If published, this will include your full peer review and any attached files.

Reviewer #2: **Yes: **Carlos Capella-Peris

---

## [Editor Report · Acceptance letter]

13 Nov 2023

PONE-D-23-12396R1 

Promoting Physical Activity and Academic Achievement through Physically Active Learning: Qualitative Perspectives of Co-Design and Implementation Processes 

Dear Dr. Simard:

I'm pleased to inform you that your manuscript has been deemed suitable for publication in PLOS ONE. Congratulations! Your manuscript is now with our production department. 

Kind regards, 

on behalf of

Dr. Josephine N. Booth 

Academic Editor

PLOS ONE